# A PON for All Seasons: Comparing Paraoxonase Enzyme Substrates, Activity and Action including the Role of PON3 in Health and Disease

**DOI:** 10.3390/antiox11030590

**Published:** 2022-03-19

**Authors:** Chrysan J. Mohammed, Sabitri Lamichhane, Jacob A. Connolly, Sophia M. Soehnlen, Fatimah K. Khalaf, Deepak Malhotra, Steven T. Haller, Dragan Isailovic, David J. Kennedy

**Affiliations:** 1Department of Medicine, University of Toledo College of Medicine and Life Sciences, Toledo, OH 43614, USA; chrysan.mohammed@utoledo.edu (C.J.M.); jacob.connolly@rockets.utoledo.edu (J.A.C.); sophia.soehnlen@rockets.utoledo.edu (S.M.S.); kareem.khalaf@utoledo.edu (F.K.K.); deepak.malhotra@utoledo.edu (D.M.); steven.haller@utoledo.edu (S.T.H.); 2Department of Chemistry and Biochemistry, University of Toledo, Toledo, OH 43606, USA; sabitri.lamichhane@rockets.utoledo.edu (S.L.); dragan.isailovic@utoledo.edu (D.I.); 3Department of Clinical Pharmacy, College of Pharmacy, University of Alkafeel, Najaf 61001, Iraq

**Keywords:** paraoxonase, substrates, lactones, cardiovascular disease, HIV, cancer

## Abstract

Paraoxonases (PONs) are a family of hydrolytic enzymes consisting of three members, PON1, PON2, and PON3, located on human chromosome 7. Identifying the physiological substrates of these enzymes is necessary for the elucidation of their biological roles and to establish their applications in the biomedical field. PON substrates are classified as organophosphates, aryl esters, and lactones based on their structure. While the established native physiological activity of PONs is its lactonase activity, the enzymes’ exact physiological substrates continue to be elucidated. All three PONs have antioxidant potential and play an important anti-atherosclerotic role in several diseases including cardiovascular diseases. PON3 is the last member of the family to be discovered and is also the least studied of the three genes. Unlike the other isoforms that have been reviewed extensively, there is a paucity of knowledge regarding PON3. Thus, the current review focuses on PON3 and summarizes the PON substrates, specific activities, kinetic parameters, and their association with cardiovascular as well as other diseases such as HIV and cancer.

## 1. Introduction

Paraoxonases (PON) are a multigene family consisting of three enzymes, PON1, PON2, and PON3, located on human chromosome 7 (7q21.3–22.1) [1,2]. These enzymes are highly conserved in mammals, having 60% and 70% homology at the amino acid and nucleotide level, respectively [1]. Human PON1 shares 66% amino acid similarity with PON2 and 61% with PON3, while human PON2 and PON3 share 66% amino acid homology. Appendix A provides a comparison of all three PON isoforms at the amino acid level including comparisons of catalytic regions as well as both predicted and known glycosylation regions. As expected, based on their sequence similarity, there is a high degree of similarity for the location of the catalytic domains between the PON isoforms and approximately 7% of the amino acid residues in each PON isoform are predicted to be glycosylated [3]. All three PONs possess high substrate specificity with overlapping but distinct substrates [4]. Recent studies have established the native physiological role of PONs to be that of a lactonase, even though they also possess arylesterase and organophosphatase activities [4,5].

All three PONs have been shown to act as antioxidants and consequently have anti-inflammatory effects in various disease states such as atherosclerosis and cardiovascular diseases [6,7,8,9]. Of the three enzymes, PON3 was the last to be discovered and is also the least studied member of the multigene family [2]. In fact, most of what is known about PON3 comes from previous research on PON1. PON3 was first identified in 1996 [10] and was discovered to be a calcium-dependent glycoprotein with a molecular weight of 40 kDa [10]. In addition to a wide range of tissue expression [11], it circulates bound to high density lipoprotein (HDL) after being synthesized in the liver (Figure 1).

Previous reviews on PONs [12,13,14] have focused mainly on the more widely studied enzyme PON1. In the current review, we focus on PON3 including its substrate specificity as well recent findings on the role of PON3 in cardiovascular diseases and other disease states such as HIV and cancer.

### 1.1. Substrates

The term “paraoxonase” originally referred to the ability of the enzyme to hydrolyze paraoxon (diethyl p-nitrophenyl phosphate) (as shown in Figure 2), which is the active metabolite of the organophosphate (OP) insecticide parathion [15,16]. Even though PON2 and PON3 are closely related to PON1, they exhibit almost no true “paraoxonase” activity (i.e., hydrolytic activity toward paraoxon as a substrate) [1,17].

Initial study of PONs related to the toxicological aspects of the enzyme because of their ability to detoxify OP compounds. More recent studies have demonstrated a correlation of PON activity with various cardiovascular diseases [13,19], diabetes [20], cancers [21], and neurological disorders [22,23].

In contrast to PON1, much less is known about PON3′s physiological role and many of the PON3 substrates identified to date are not endogenous biological compounds. Although some potential biological substrates have been discovered, the true physiological substrate for this enzyme is still enigmatic. More research on PON3 with the aim of discovering new endogenous substrates is necessary in order to elucidate its physiological significance and to determine diagnostic and/or therapeutic potential.

The substrates for PONs’ hydrolytic activity are generally classified into three categories based on their molecular structure as follows: organophosphates, aryl esters, and lactones (Figure 3 and Table 1 summarizes the substrates and their activities).

#### 1.1.1. Organophosphates (OP)

OPs are commonly used insecticides. They are phosphate triesters with a general structure of O = P(OR)_3_. Their mechanism of action is associated with the inhibition of acetylcholinesterase in the nervous system, which catalyzes the hydrolysis of the neurotransmitter acetylcholine, resulting in acute cholinergic crisis [24]. Except for poorly hydrolyzing paraoxon, PON3 is almost devoid of OP activity [4]. PON2 also demonstrates almost no paraoxonase activity [4]. However, PON1 strongly hydrolyzes the active metabolites of several OP insecticides (parathion, chlorpyrifos, malathion, and diazinon) [25] and nerve agents (tabun, sarin, soman, and cyclosarin) [18,26]. Paraoxon is one of the best-known substrates for PON1 and there is great potential for PON1 to be used therapeutically as an antidote for acute OP poisoning.

#### 1.1.2. Aryl Esters

Structurally, aryl esters are esters that have an aromatic ring. While all three PON isoforms exhibit arylesterase activity, PON2 has very low arylesterase activity [27,28]. Phenyl acetate is one of the best-known aryl ester substrates for PON1 but is hydrolyzed at a modest rate by PON3 and very slowly by PON2 [4,5]. The rate of hydrolysis of aromatic esters by PONs are in order: PON3 >> PON1 > PON2 [27]. PON3′s active site has a preference for bulky substrates such as statin lactone (cyclic ester) drugs [4], which is possibly why PON3 demonstrates higher hydrolytic activity toward bulky steroid structures possessing estrogen esters compared to PON1.

#### 1.1.3. Lactones

Lactones are cyclic carboxylic esters, containing an ester bond (−C(=O)−O−). Phylogenetic, structural, and biochemical data revealed that all three PONs are in fact lactonases/lactonizing enzymes with overlapping but distinct substrate specificity while arylesterase and phosphotriesterase are promiscuous activities of these enzymes [4,28,29,30,31,32,33]. Lactones have the ability to affect cellular growth, signaling, and differentiation. Therefore, it is probable that PONs exert their physiological functions by metabolizing and thus altering the biological activity of endogenous and exogenous lactones and can play an important role in health and diseases [29]. PONs hydrolyze lactones far more quickly than their respective noncyclic (or “open”) ester analogs [25]. Further structure–activity studies have demonstrated that lipophilic five and six-membered lactone rings (i.e., γ- and δ-lactones) are hydrolyzed by all three PONs with high specificity, suggesting that these compounds are representative of native substrates [30,31].

The lactonase activity of PON2 is much more limited [16,32]. Compared to PON1 and PON3, PON2 exhibits the highest specific activity for the hydrolysis and inactivation of *N*-acylhomoserine γ-lactones (HSLs), which are bacterial quorum sensing molecules that act as a defense mechanism against bacterial pathogens [16,24]. The hydrolytic activities of PONs for 3-oxo-C12-HSL, a signaling molecule that regulates pro-inflammatory and apoptosis gene expression, were in the order: PON2 >> PON1 > PON3 [33]. N-acyl-HSLs are established as the native substrates of PON2.

All PONs play a major role in the hydrolysis, and consequently, detoxification of homocysteine thiolactone (HCTL). Increased plasma levels of HCTL is a risk factor for atherosclerotic vascular diseases [34,35]. Additionally, the non-physiologic aromatic lactones, dihydrocoumarin and 2-coumaronone, are good substrates for PON3 and PON1 [36].

Although the crystal structure of PON3 is yet to be determined, based on substrate specificity experiments using purified recombinant PONs, it appears that the active site of PON3 is larger than that of PON2 and PON1 [4]. PON3, therefore, has a greater ability to hold and hydrolyze bulky substrates, whereas PON1 is able to catalyze non-substituted and short chain-substituted lactones [5,28]. Interestingly, PON3 exclusively hydrolyzes the bulky cardiovascular drugs spironolactone and statin lactones such as mevastatin, lovastatin, and simvastatin. Lovastatin has been used to assess the selective enzymatic activity of PON3 in tissue as well as cell lysates [5]. Simvastatin effects are exerted by hydrolysis to simvastatin acid, and paraoxonases, along with carboxylesterases, are credited for this activation. Simvastatin was also found to reduce atherosclerosis in patients with advance chronic kidney disease (CKD) [37,38]. Moreover, the renin–angiotenin–aldosterone system (RAAS) is an important target in pharmacotherapy in patients that suffer cardiovascular and renal diseases. Spironolactone is a common drug used to inhibit the downstream effects of RAAS and data suggest positive benefits for heart failure patients suffering from CKD [39].

Endogenous cardiotonic steroids (CTS) such as telocinobufagin and marinobufagenin, belonging to the family of bufadienolides, are bulkier compounds and are similar to bulky statin drugs such as lovastatin in having a six-membered lactone ring in their structure [40], CTS may therefore be putative PON substrates. CTS are important in sodium homeostasis at physiologic levels [41]. The biological effects of elevated CTS in volume-expanded conditions such as preeclampsia [42], CKD [43,44], and end-stage renal failure [45] as well as on various organs is well established [46]. It is appealing to study whether PONs via their lactonase activity can hydrolyze the closed lactone ring of CTS into the open acid form. Such studies will be useful in establishing CTS as a new potential substrate for PONs.

### 1.2. PONs and Bioactive Lipids of Arachidonic Acid

Arachidonic acid and its active derivatives eicosanoids have attracted more attention as PON substrates in recent years. These compounds mediate a number of metabolic processes in vivo and are involved in the development of various cardiovascular and renal diseases [47]. All three PONs are able to metabolize, with very high-efficiency, 5-hydroxy-eicosatetraenoic acid 1,5-lactone (5-HETEL) and 4-hydroxy-docosahexaenoic acid (4-HDoHE), products of both the enzymatic and non-enzymatic oxidation of arachidonic acid and docosahexaenoic acid, respectively [5,48]. Notably, 5-HETEL was found to be the best substrate for all three PONs identified thus far [32]. PON1 has the highest activity followed by PON3 and PON2 having very little activity toward this substrate. 5-HETE and 5-HETEL are known to inhibit platelet neutrophil phospholipase A2 (PLA2) along with peritoneal macrophage cyclooxygenase (COX) [49]. The ability of paraoxonase to hydrolyze 5-HETEL indicates a possible physiological mechanism of regulation.

PON3 has by far the greatest hydrolytic efficacy toward eicosanoids, 5,6-dihydroxy-eicosatrienoic acid lactone (5,6-DHETL), and cyclo-epoxycyclopentenone (cyclo-EC), followed by PON1 and again with PON2 having little or no activity toward these substrates. 5,6-Epoxyeicosatrienoic acid (5,6-EET) is spontaneously lactonized into 5,6 DHETL (Figure 4). PON3 efficiently hydrolyzed 5,6-DHETL to its corresponding 5,6-dihydroxytrienoic acid (5,6-DiHET) with a specific activity of approximately 15-fold higher than PON1 [32]. In vivo studies demonstrated that in canine coronary microcirculation 5,6 DHETL is an extremely potent vasodilator [50] and scant intrarenal availability of dihydroxyeicosatrienoic acids aided the progression of CKD and end-stage renal disease development [51]. Epoxycyclopentenone (EC) can readily undergo lactonization to cyclo-EC at physiological pH, which is in turn hydrolyzed to dihydroxy-EC by paraoxonase [32,52]. Both cyclo-EC and its dihydroxy hydrolysis product were shown to exhibit potent anti-inflammatory activities although the lactone was found to be more potent, as investigated by in vitro and in vivo studies [32,53].

This biologically active δ-lactone with unique biological activities is likely a native endogenous substrate for PONs. However, the physiological relevance of their metabolism by PONs remains elusive. More studies are needed to establish eicosanoids as concrete physiological substrate of PON3 and to clearly elucidate their metabolite’s physiological significance.

### 1.3. PON Activity Assays

The measurement of PONs is indirectly carried out by evaluating their enzymatic activities toward different substrates, which is examined most commonly by spectrophotometric assays [54]. PON1 can be determined by its paraoxonase activity (using paraoxon as a substrate) [55,56,57], arlyesterase activity (using phenyl acetate as a substrate) [58], and lactonase activity (using several lactones such as dihydrocoumarin as a substrate) [5]. PON2 is commonly measured by using acylhomoserine lactone as a substrate [59]. PON3 lactonase activity is mainly determined by the hydrolysis of statin lactones (simvastatin, lovastatin, mevastatin) [4]. A study was conducted to measure PON3 hydrolysis of simvastatin in human blood serum. Simvastatin (substrate) and simvastatin acid (product) in the reaction mixture was determined by high-performance liquid chromatography (HPLC) with an ultraviolet absorbance detection at a wavelength of 239 nm [60]. Since there was no difference between the spectra of the substrate and the product, HPLC was used for their separation as the substrate and product elute at different retention times. In another study [61], the HPLC method was used for the determination of lovastatinase activity in the livers of human PON3 transgenic male mice. The amount of hydrolysis product generated was semi-quantitatively measured. It was demonstrated that an increase in PON3 activity significantly decreased atherosclerotic lesion formation and adiposity. A study conducted by Teiber et al. determined PON3 catalyzed hydrolysis of 5,6-DHTL and cyclo-EC. Analysis was carried out using HPLC equipped with a UV/Vis detector, set at 254 nm for cyclo-EC and 205 nm for 5,6-DHTL. They calculated the specific activities of recombinant PONs for these eicosanoid lactones [32]. Specific activities of PONs towards various substrates are summarized in Table 2 and kinetic measurement data for various substrates of PONs is presented in Table 3.

### 1.4. PON Concentration Assays

Serum samples are mostly used for the direct quantification of PON enzymes and to establish the correlation of the PON level with various diseases. Marie-Claudie et al., in 1994, first reported the development of a highly specific enzyme-linked immunosorbent assay (ELISA) for the quantification of human serum PON1 concentration using a previously characterized monoclonal antibody [56]. Various immunoassay based methods utilizing PON antibodies have been extensively used to measure PON concentrations [76]. PON2 is an intracellular enzyme that is located in the membranes of cell organelles and is not detected in serum under normal physiological conditions [9]. PON3 quantification has been hindered by the lack of reliable methods to measure its levels in the circulation. PON3 measurement was not conducted until 2011, when Aragones et al. employed a novel in-house ELISA using polyclonal antibodies specific for PON3 to determine the PON3 concentration in patients with chronic liver disease and in controls. They generated an anti-PON3 antibody by inoculating rabbits with a synthetic peptide specific to mature PON3. This antibody was used to develop an ELISA. They reported the median serum paraoxonase activity in the control population to be 142.9 U/mL, which was significantly lower in the group with type 1 diabetes (124.1 U/mL) and in the group with type 2 diabetes (123.4 U/mL) [77]. In a study conducted by Judith et al., PON3 measurement in HDL was carried out using liquid chromatography tandem mass spectrometry (LC-MS/MS), which was further confirmed by western blot analysis. They found that the PON3 level was depleted from the HDL of autoimmune disease patients with subclinical atherosclerosis [78].

### 1.5. PON3 in Cardiovascular Disease

Cardiovascular diseases (CVDs) are responsible for the largest portion of mortalities worldwide with coronary artery disease (CAD) being the largest contributor [79,80]. Increased oxidant stress is responsible for numerous inflammatory processes that contribute to the underlying pathogenesis of cardiovascular diseases including atherosclerosis and CAD. The endothelium is a monolayer essential for regulating vascular homeostasis, vascular tone, and inflammation of vessel walls [81]. Endothelial dysfunction is a hallmark of CVD and its primary cause is an imbalance between antioxidant defense systems and generation of reactive oxygen species [82,83]. Oxidative stress affects endothelial cell viability and permeability by oxidizing cellular components and inducing inflammation [84,85]. Numerous studies have demonstrated that pharmacological regulation of oxidative stress and inflammation can prevent endothelial dysfunction and have potential for CVD treatment [82,86].

All three PONs possess well-established antioxidant and anti-inflammatory functions [6,87,88,89,90]. Priyanka et al. have extensively reviewed the role of PON3 in CAD [7]. Studies involving mouse models have provided clear evidence for the antiatherogenic role of the PON genes [61,91,92,93,94,95,96,97]. The PON gene family and atherosclerosis related cardiovascular disease [20] and more specifically, the role of PON3 in atherosclerosis, have been thoroughly reviewed [98]. Briefly, the underlying mechanism of PON3′s atheroprotection is that it locates to the mitochondrial membrane where it binds to ubisemiquinone, the electron donor for superoxide formation, inhibiting the generation of superoxide, resulting in reduced inflammation [99]. Diminished superoxide production consequently prevents atherosclerotic plaque development since uptake of oxLDL by macrophages forms the basis for plaque development.

Atherosclerosis is the underlying causative disorder for peripheral artery disease (PAD) [100]. Rull et al. demonstrated that serum PON3 levels were significantly increased in patients with PAD compared to healthy controls [101]. There was also a positive correlation between PON3 and insulin levels and the homeostasis model assessment index. An earlier study by Shih et al. reported that human PON3 transgenic mice had lower insulin compared to controls [61]. These findings suggest a potential role for PON3 in regulating glucose metabolism.

Patients with diabetes have an increased risk of developing CVD, with research pointing to factors that increase the risk of atherosclerosis [102,103]. Apolipoprotein AI is a lipoprotein that is necessary for maintaining an operable environment for enzymes such as PONs [104]. PON3′s lactonase activity is augmented by binding to apoA-I [73]. In patients with type 2 diabetes mellitus (T2DM), the severity of CAD is associated with an increase in apolA-I glycation and decrease in HDL-associated PON1 and PON3 activity. Furthermore, PON3 is depleted from HDL in autoimmune disease patients with atherosclerosis. HDL associated PON prevents the oxidation of HDL and maintains its functions [105], therefore, these findings support that dysfunctional HDL associated PON poses a cardiovascular risk [78].

Dysfunctional HDL and dyslipidemia are common abnormalities among chronic kidney disease (CKD) patients, which contribute to promoting inflammation and oxidative stress that is prevalent in the milieu of this disease. These abnormalities increase the risk of cardiovascular disease in CKD and patients with CKD suffer significant cardiovascular mortality and morbidity. Recent studies have highlighted a decrease in circulating PON lactonase activity across CKD etiologies, which is associated with lower HDL level, and is predictive of increased major adverse cardiac events in these patients [106,107]. Based on these findings, enhancing PON lactonase activity in CKD may have the potential to reduce cardiovascular events in CKD. However, future studies are required to test this hypothesis. While there are several studies highlighting the relevance of PON1 and total circulating PON in CKD, there have been no studies focusing on the significance of PON3 specifically in CKD.

### 1.6. PON3 in Other Disease States

There is an emerging interest regarding the role of PON2 and PON3 in cancer due to their remarkable upregulation in some tumor tissues [104]. For the purpose of this review, we focused on PON3. In models of multi-drug resistant esophageal cancer, in vitro studies showed that PON3 is hypermethylated at the promoter region, which consequently downregulates its expression [108]. Further in vitro studies using the K150 cell line transfected with GFP-PON3 revealed that PON3 suppressed migration and invasion of esophageal cancer cells. In this same study, in vivo findings demonstrated that PON3 inhibited tumor growth and drug resistance [108]. PON3 hypermethylation could also serve as a mechanism for explaining the association between smoking and adverse prostate cancer outcomes. Shui et al. (2016) found that hypermethylation in the promoter region of the PON3 gene was associated with smoking status and strongly correlated with mRNA expression in a cohort of 523 men who underwent radical prostatectomy [109].

Furthermore, studies of the relationship between PON3 and hepatocellular carcinoma (HCC) have demonstrated that PON3 inhibits cell proliferation, and downregulation of PON3 contributes to HCC disease progression [110,111]. Analysis of publicly available GEO datasets, microarrays, immunohistochemistry, and qRT-PCR revealed that PON3 expression is significantly decreased in HCC tissues [110,111]. These studies highlighted the clinical implications of PON3′s downregulation since low levels of PON3 predicted decreased recurrence-free survival and overall survival and is an independent risk factor for overall survival and time to recurrence in patients with HCC [110,111]. PON3 expression level negatively correlates with tumor size and number, suggesting that PON3 expression may inhibit HCC cell proliferation [110]. Findings from Jin et al. and Cia et al. agreed that the biological function of PON3 in HCC is tumor-suppressive. These results indicate that PON3 may serve as a prognostic marker in HCC.

While studies have shown that PON3′s expression serves to inhibit cell proliferation through interference in cell cycle progression, invasion, and migration in certain cancers, it has also been shown to display oncogenic properties [105]. The contrast in the findings suggest that PON3 may serve as a double-edged sword in disease progression due to differences in regulation of its enzymatic function as an antioxidant in various cancer pathways [98]. More research is needed in exploring these differences in cancer. Schweikert et al. (2012) showed that the same protective effect of PON3 against obesity and atherosclerosis due to its antioxidant property also defines anti-apoptotic and oncogenic roles in human cancers [99]. PON3 directly acts at mitochondrial membranes to inhibit normal apoptotic activities. PON3 is overexpressed in a variety of human cancer types and is shown to abrogate mitochondrial superoxide production, consequently granting anti-oxidative and anti-apoptotic benefits, allowing tumor cells to escape death [99]. The mechanism by which PON3 regulates cell death is extensively reviewed by Witte et al. [98].

PON3 levels vary between different types of cancer. A meta-analysis of PON3 showed that PON3 is downregulated not only in HCC, but also in clear cell sarcomas of the kidney, ovarian serous papillary carcinomas, cervical carcinomas, papillary thyroid carcinomas, prostate carcinomas, and non-Hodgkin’s lymphoma while PON3 was upregulated in lung adenocarcinoma and pancreatic carcinoma [98]. Furthermore, in oral squamous cell carcinoma (OSCC), PON3 was identified to be upregulated via the PI3K/Akt pathway. Inhibition of PON3 in human OSCC cell lines, TSC-15 and CAL27, significantly reduced cell proliferation and metastasis in vitro. Similarly, when the cell lines were transfected with PON3 siRNA and transplanted into nude mice, the weight and volume of OSCC tumors decreased significantly, demonstrating that the inhibition of PON3 reduced OSCC progression [112].

Because patients with human immunodeficiency virus (HIV) tend to develop metabolic complications associated with atherosclerosis and concomitant CAD, PON3 has been of interest in studying the course of this disease. Serum PON3 concentration was three times higher in HIV-infected patients and inversely related to oxidized LDL levels. Long-term use of nonnucleoside reverse transcriptase inhibitor (NNRTI) based antiretroviral therapy was associated with a decrease in serum PON3 concentrations. Taken together, these results suggest that antiretroviral treatments may induce oxidative stress through downregulation of PON3 and provide evidence for an anti-oxidative protective role of PON3 in the setting of HIV [77]. Later studies using lipid profiling of HDL from HIV-infected and uninfected controls revealed that HDL particles from patients with HIV have reduced PON3 levels and PON activity [113]. The decreased activity of this atheroprotective enzyme provides insight into the lipoprotein derangements that are experienced in the HIV population. However, further studies are needed to determine whether PON3 is mechanistically linked to lipoprotein abnormalities in patients with HIV.

PON3 has also been shown to have a hepatoprotective role that prevents histological changes and liver cell apoptosis, which leads to liver disease [114]. Furthermore, according to Peng et al., the delivery of human PON3 to mice with carbon tetrachloride (CCl4) induced liver injury led to an improved liver histological architecture, indicating that PON3 plays a role in protecting the liver from injury [115]. This hepatoprotective effect of PON3 is closely related to its lactonase and antioxidant activities [116]. Furthermore, PON3 expression was also significantly reduced in the liver of CCl4 treated rats, indicating a connection between PON3 and the prevention of liver injury [116]. The protective role of PON3 in liver disease is expected since paraoxonases have a protective role against oxidative stress, which plays an important role in the pathogenesis of liver disease [117].

## 2. Conclusions

Of the three members of the paraoxonase family, PON3 was the last to be discovered and is the least studied. While named for their ability to hydrolyze paraoxon, PON2 and PON3 showed almost no paraoxonase activity compared to PON1. PON3 also had low arylesterase activity compared to PON1 with PON2 having the least activity. Furthermore, all three PONs had lactonizing activity and it is probable that the PONs exert their physiological functions by altering the activity of lactones. The ability of all three PONs to hydrolyze five- and six-membered lactone rings suggests that these compounds are representative of the native substrates of PONs. Endogenous cardiotonic steroids containing six-membered lactone rings are elevated in volume-expanded conditions and may serve as novel substates for PONs. Eicosanoids have also attracted interest as PON substrates, specifically 5-HETE and 5-HETEL, which are known to inhibit PLA2 and COX activity. The role of PON3 in atherosclerosis has been previously reviewed but there is increasing interest in the role of PON3 in cancer. Studies have shown that PON3 may serve as a double-edged sword when it comes to cancer as it is downregulated in some cancer types, but the antioxidant effect of PON3 helps to inhibit normal apoptotic mechanisms in others. Furthermore, long-term use of NNRTIs to treat HIV is associated with a decrease in serum PON3 concentrations, indicating that antiretroviral treatments may induce oxidative stress through the downregulation of PON3. Finally, while potential substrates for PON3 have been discovered, more studies are necessary to determine its true physiological substrate.

## Figures and Tables

**Figure 1 antioxidants-11-00590-f001:**
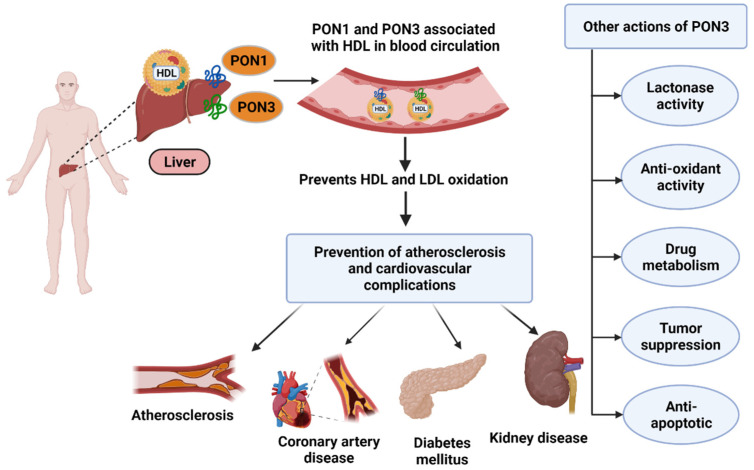
PON synthesis in liver and their roles in various diseases (created with Biorender.com (Date when last accessed: 21 February 2022).

**Figure 2 antioxidants-11-00590-f002:**
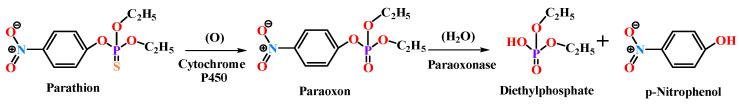
Paraoxon hydrolysis reaction for which paraoxonase derives its name [18].

**Figure 3 antioxidants-11-00590-f003:**
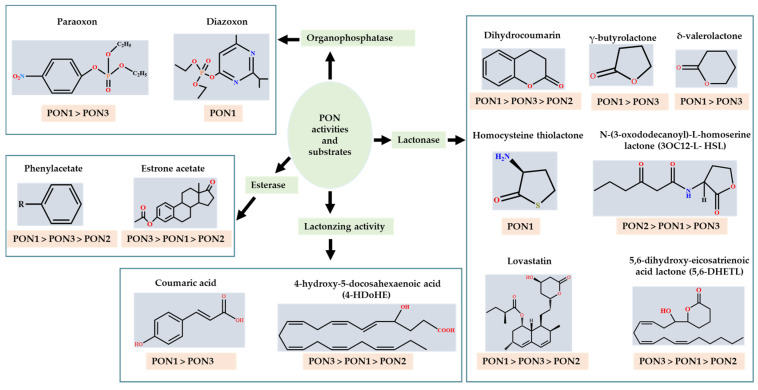
Activities of PONs and their substrates.

**Figure 4 antioxidants-11-00590-f004:**
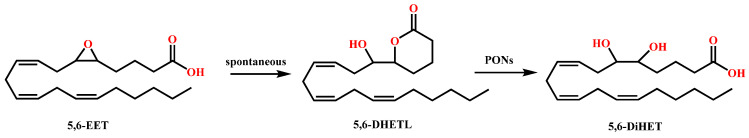
Hydrolysis of 5,6-DHETL by PONs [32].

**Table 1 antioxidants-11-00590-t001:** Summary of PON’s substrate activities.

Activity	PON1	PON2	PON3
Organophosphate	Yes [18]	No [4]	No (except paraoxonase) [4]
Arylesterase	Yes [25,27]	Yes, but very low [4,27]	Yes, but mostly low [5]
Lactonase	Yes [25]	Yes [33]	Yes, high [36]
Eicosanoid	Yes [4]	Little or no [4]	Yes [4,32]

**Table 2 antioxidants-11-00590-t002:** Specific enzyme activities of the PONs.

Substrate	PON1	PON2	PON3	Source	Ref.	Biological Role
Organophosphate (μmol/min/mg)				Purified recombinant human PONs	[4]	Inhibition of acetylcholinesterase, leading to cholinergic syndrome [24]
Paraoxon (1 mM)	1.94	ND	0.205			
Chlorpyrifos oxon (0.32 mM)	40.9	ND	ND			
Diazoxon (1 mM)	113	ND	ND			
Esters (μmol/min/mg)				Purified recombinant human PONs	[4]	Phenyl acetate can be used as anticancer drug [62]
Phenyl acetate (1 mM)	1120	0.086	4.1			
p-NO_2_-acetate (1 mM)	15.0	0.7	39.0			
p-NO_2_-propionate (1 mM)	13.6	0.96	20.7			
p-NO_2_-butyrate (1 mM)	1.3	1.4	11.4			
Thiophenyl acetate (1 mM)	259	ND	0.48	Purified rabbit serum	[5]	
β-Napthyl acetate (0.5 mM)	139	ND	4.6			
Estrone acetate (25 µM)	0.137	0.004	0.515	Purified recombinant human PONs	[27]	Estrogens are antioxidants [27]
Estrone propionate (25 µM)	0.057	ND	0.220			
17β-Estradiol acetate (25 µM)	0.125	0.005	1.06			
17β-Estradiol diacetate (10 µM)	1.00	0.509	55.4			
Estrone Enol diacetate (10 µM)	0.987	0.955	31.8			
17β-Estradiol 3-Ac 17-cyclopentyl-propionate (10 µM)	0.024	0.114	1.60			
Aromatic lactones (μmol/min/mg)				Purified recombinant human PONs	[4]	3,4 DHC exhibits neuroprotective activity [63] Homogentisic acid lactone prevents infection against *Pseudomonas Aeruginosa* [64]
Dihydrocoumarin (1 mM)	129.9	3.1	126.1			
2-Coumaronone (1 mM)	135.7	10.9	40.7			
Homogentisic acid lactone (HgAL) (1 mM)	329.5	ND	ND			
γ-lactones (μmol/min/mg)				Human recombinant PON	[4]	
γ-Butyrolactone (1 mM)	32.1	ND	0.81			
γ-Valerolactone (1 mM)	45.0	ND	6.2			
γ-Hexalactone (1 mM)	51.7	ND	23.9			
γ-Heptalactone (1 mM)	57.2	ND	27.7			
γ-Octalactone (1 mM)	69.2	ND	25.6			
γ-Nonalactone (1 mM)	144.7	ND	30.9			
γ-Decanolactone (1 mM)	173.8	ND	45.6			
γ-Undecanolactone (1 mM)	127.6	ND	71.4			
α-Angelica lactone (1 mM)	183.0	ND	20.7			
γ-Phenyl-γ-butyrolactone (0.5 mM)	63.0	0.68	11.4			
δ-lactones (μmol/min/mg)						
δ-Valerolactone (1 mM)	671	ND	14.5			
δ-Hexalactone (1 mM)	72	ND	11.7			
δ-Nonalactone (1 mM)	150	ND	11.1			
δ-Decanolactone (1 mM)	251	ND	44.3			
δ-Undecanolactone (1 mM)	287	ND	84.4			
δ-Tetradecanolactone (0.5 mM)	154	ND	22.7			
DL-3-Oxo-hexanoyl-HSL (250 µM)	0.0334	0.2683	ND			
L-3-Oxo-hexanoyl-HSL (250 µM)	N	0.5080	ND			
DL-Heptanoyl-HSL (25 µM)	0.0036	0.0311	0.0049			
DL-Dodecanoyl-HSL (25 µM)	0.0167	0.4588	0.0877			
DL-Tetradecanoyl-HSL (25 µM)	0.0035	0.4239	0.0255			
Eicosanoid lactones (µmol/min/mg)				Human recombinant PON ^a^ Recombinant PON	^a^ [4] [32]	Control homeostatic and inflammatory processes, potent vasodilators [50]
5-HETEL (10 µM)	75.4 ^a^	1.83 ^a^	27.5 ^a^			
	(192_Q_)					
	(192_R_)					
Cyclo-EC	0.007 0.008	<0.002	19.2			
(±)−5 (6)-DHETL	0.62 0.67	0.67	12.7			
Homoserine lactones (nmol/min/mg)	(192_Q_) (192_R_)			Purified human PON	[33]	Prevents bacterial infection [65]

5-HL (10 µM)	29,500					
3OC12-L-HSL (10 µM)	36,800	3100	22,100			
	224 334	7647	100			
Amino acid derived lactones (µmol/min/mg) Homocysteine thiolactone (10 mM)	(192_Q_) (192_R_) 0.02 0.03	ND	ND	Purified human PON1 Q and R	[31]	Risk factor for atherosclerosis [66]
Lactone drugs (pmol/min/mg)	(192_Q_) ^b^ (192_R_) ^b^			Purified human PON1 Q and R ^b^ Human recombinant PON ^c^	^b^ [31] ^c^ [4]	Lipid lowering drugs; used as cardiovascular medicine [67]
Mevastatin (12.8 µM) ^b^	485.1 461.8	ND	ND			
Lovastatin (12.8 µM) ^b^ (25 µM) ^c^	489.6 473.4	ND	(2.66 × 10^5^) ^c^			
Simvastatin (12.4 µM) ^b^ (25 µM) ^c^	684.5 568.3	ND	(1.1 × 10^4^) ^c^			
Spironolactone (12.0 µM) ^b^ (25 µM) ^c^	234.8 262.9	ND	(1.3 × 10^4^) ^c^			
Lactonizing activity (μmol/min/mg)				Recombinant PON	[4]	p-coumaric acid has antioxidant, and anti-inflammatory properties [68] 4-HDoHE triggers inflammation [69]
Coumaric acid (100 µM)	0.047	ND	0.013			
4-HDoHE (10 µM)	1.51	0.52	13.7			

Superscripts (^a–c^) are used to indicate the references from which the values for specific enzyme activities for each PONs were obtained.

**Table 3 antioxidants-11-00590-t003:** Kinetic parameters for substrate hydrolysis by PON1 and PON3.

Substrate	Structure	Kinetic Constants	PON1	PON3	Source	Ref.
Paraoxon	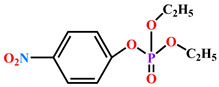	k_cat_ (s^−1^)	126 ^a^	0.001 ^b^	rePON1G2E6 ^a^ RabPON3 ^b^	^a^ [70] ^b^ [71]
		K_M_ (mM)	0.9 ^a^	1.3 ^b^		
		k_cat_/K_M_ (s^−1^ M^−1^)	1.4 × 10^5 a^	0.73 ^b^		
Chlorpyrifos oxon	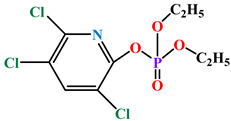	k_cat_ (s^−1^)	7800	ND	PON1Q192	[72]
		K_M_ (mM)	0.075	ND		
		k_cat_/K_M_ (s^−1^ M^−1^)	1.0 × 10^8^	ND		
Sarin	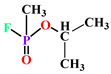	k_cat_ (s^−1^)	190	ND	PON1Q192	[31]
		K_M_ (mM)	0.21	ND		
		k_cat_/K_M_ (s^−1^ M^−1^)	9.0 × 10^5^	ND		
Soman	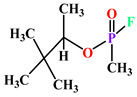	k_cat_ (s^−1^)	1150	ND	PON1Q192	[31]
		K_M_ (mM)	0.42	ND		
		k_cat_/K_M_ (s^−1^ M^−1^)	2.7 × 10^6^	ND		
2-napthylacetate	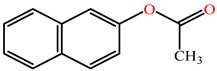	k_cat_ (s^−1^)	ND	0.66	RabPON3	[71]
		K_M_ (mM)	ND	0.211		
		k_cat_/K_M_ (s^−1^ M^−1^)	ND	3100		
Phenylacetate	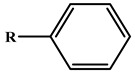	k_cat_ (s^−1^)	698	ND	rePON1G2E6	[25]
		K_M_ (mM)	1.2	ND		
		k_cat_/K_M_ (s^−1^ M^−1^)	5.95 × 10^5^	ND		
5-thiobutil butyrolactone	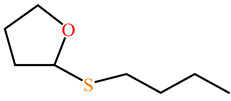	k_cat_ (s^−1^)	116	42	PON1 variant P2E6, rePON3	[73]
		K_M_ (mM)	0.27	0.44		
		k_cat_/K_M_ (s^−1^ M^−1^)	4.4 × 10^5^	99,200		
γ-nonanoic lactone	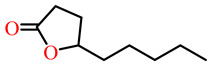	k_cat_ (s^−1^)	31	22	PON1 variant P2E6, rePON3	[73]
		K_M_ (mM)	0.39	1.1		
		k_cat_/K_M_ (s^−1^ M^−1^)	7.8 × 10^4^	20,550		
γ-undecanoic lactone	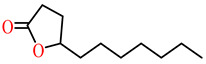	k_cat_ (s^−1^)	62	21	PON1 variant P2E6, rePON3	[73]
		K_M_ (mM)	0.60	0.47		
		k_cat_/K_M_ (s^−1^ M^−1^)	1.03 × 10^5^	43,700		
δ-undecanoic lactone	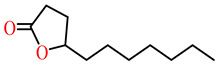	k_cat_ (s^−1^)	ND	21	PON1 variant P2E6, rePON3	[73]
		K_M_ (mM)	ND	0.8		
		k_cat_/K_M_ (s^−1^ M^−1^)	ND	28,700		
γ-butyrolactone	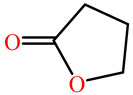	k_cat_ (s^−1^)	147.209	ND	G3C9 PON1 wild Type	[74]
		K_M_ (mM)	36.800	ND		
		k_cat_/K_M_ (s^−1^ M^−1^)	4 × 10^3^	ND		
γ -valerolactone	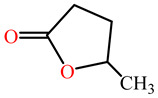	k_cat_ (s^−1^)	36.621	ND	G3C9 PON1 wild Type	[74]
		K_M_ (mM)	1.316	ND		
		k_cat_/K_M_ (s^−1^ M^−1^)	2.78 × 10^4^	ND		
Dihydrocoumarin	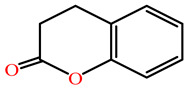	k_cat_ (s^−1^)	152 ^c^	1321 ^d^	rePON1	^c^ [25] ^d^ [75]
		K_M_ (mM)	0.129 ^c^	0.75 ^d^	G2E6 ^c^	
		k_cat_/K_M_ (s^−1^ M^−1^)	1.19 × 10^6 c^	1761 ^d^	Purified rat PON ^d^	

Superscripts (^a–d^) are used to indicate the references from which the values for specific enzyme activities for each PONs were obtained.

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
