# Peer review of "A PON for All Seasons: Comparing Paraoxonase Enzyme Substrates, Activity and Action including the Role of PON3 in Health and Disease"

_antioxidants, 2022, doi:10.3390/antiox11030590_

Round 1

Reviewer 1 Report

The manuscript “A PON for All Seasons: Comparing paraoxonase enzyme substrates, activity and action in health and disease” is a review article that summarizes the PONs substrates, specific activities, kinetic parameters and their association with cardiovascular as well as other diseases. The manuscript is generally well written and easy to read.

However, authors should address some points before the manuscript could be considered for publication.

Title: the title “A PON for All Seasons: Comparing paraoxonase enzyme substrates, activity and action in health and disease” is not enough specific, since the health and disease conditions discussed in this manuscript are related only to PON3. I suggest authors to change the title in “A PON for All Seasons: Comparing paraoxonase enzyme substrates, activity and action and role of PON3 in health and disease”.

Line 213: authors should specify that PON2 is an intracellular enzyme that locates in the membranes of cell organelles.

Line 229: “Increased oxidant stress is responsible for numerous inflammatory processes that contribute to the underlying pathogenesis of cardiovascular diseases, including atherosclerosis and CAD”. Authors should expand this section by adding more information about the toxicity linked to ROS overproduction. Indeed, oxidative stress heavily impact viability and permeability of endothelial cells (PMID: 34153425; PMID: 33123312), oxidizing cellular components and inducing inflammation.

The names PON2 and PON3 are sometimes written as PON-2 and PON-3 (e.g. line 273). Please reconcile throughout the text using only PON2 and PON3.

Line 273: Since authors state that “There is an emerging interest regarding the role of PON2 and PON3 in cancer due to their remarkable upregulation in some tumor tissues”, references to justify this statement are needed (PMID: 33210737; PMID: 34302630).

Authors should discuss more about the double-edged sword PON3 impact on cancer.

Author Response

REVIEWER 1:

The manuscript “A PON for All Seasons: Comparing paraoxonase enzyme substrates, activity and action in health and disease” is a review article that summarizes the PONs substrates, specific activities, kinetic parameters and their association with cardiovascular as well as other diseases. The manuscript is generally well written and easy to read.

However, authors should address some points before the manuscript could be considered for publication.

  • Title: the title “A PON for All Seasons: Comparing paraoxonase enzyme substrates, activity and action in health and disease” is not enough specific, since the health and disease conditions discussed in this manuscript are related only to PON3. I suggest authors to change the title in “A PON for All Seasons: Comparing paraoxonase enzyme substrates, activity and action and role of PON3 in health and disease”.

Reply: Thank you for this suggestion, we agree and have now changed the title accordingly.

  • Line 213: authors should specify that PON2 is an intracellular enzyme that locates in the membranes of cell organelles.

Reply: Thank you for this clarification, we agree and have now changed Line 213 ( 472 in revised manuscript) accordingly to reflect the localization of PON2.

  • Line 229: “Increased oxidant stress is responsible for numerous inflammatory processes that contribute to the underlying pathogenesis of cardiovascular diseases, including atherosclerosis and CAD”. Authors should expand this section by adding more information about the toxicity linked to ROS overproduction. Indeed, oxidative stress heavily impact viability and permeability of endothelial cells (PMID: 34153425; PMID: 33123312), oxidizing cellular components and inducing inflammation.

Reply: Thank you for this suggestion, we agree and have now expanded the section around Line 229 to discuss toxicity linked to ROS overproduction including its impact on viability and permeability of endothelial cells, oxidizing cellular components and inducing inflammation. We have included new references for this section including those suggested (PMID: 34153425; PMID: 33123312) as well as additional references of pertinent reviews in this area 

  • The names PON2 and PON3 are sometimes written as PON-2 and PON-3 (e.g. line 273). Please reconcile throughout the text using only PON2 and PON3.

Reply: Thank you for this clarification, we agree and have now reconciled these references to PON's throughout the manuscript.

  • Line 273: Since authors state that “There is an emerging interest regarding the role of PON2 and PON3 in cancer due to their remarkable upregulation in some tumor tissues”, references to justify this statement are needed (PMID: 33210737; PMID: 34302630).

Reply: Thank you for this suggestion, we agree and have now included new references for this section including those suggested (PMID: 33210737; PMID: 34302630).

  • Authors should discuss more about the double-edged sword PON3 impact on cancer.

Reply: Thank you for this suggestion, we agree and have now included additional discussion of the double-edged role of PON3 on cancer and included additional references 

Reviewer 2 Report

PONs are an essential group of enzymes involved in the metabolism of various substrates, including organophosphates, aryl esters, and some lactones. In this review, the authors focus mainly on PON3 members of paraoxonases and describe their specific activities and their role in various diseases such as cancers, HIV, and CVD. The topic is generally described in detail and seems technically sound, and the results are of general interest. The manuscript is, in principle, suitable for publication if the issues below are adequately addressed in a revised version.
MAJOR comment
Because the review may be of general interest, I suggest comparing PONs members at a molecular level (amino acid alignments, domain composition, etc.). Are there any differences between the glycosylation profile of paraoxonases among PON1-3?

MINOR suggestions
Page 2, line 51: the sentence: “…they exhibit almost no true “paraoxonase” activity…” is unclear. Could the authors explain that in a more detailed manner?
Page 2, subsection 1.1.2: there is confusion, as the authors indicate that PON3 has low arylesterase activity compared to PON1 (lines 80-81), however in line 84, they tell something different.
Page 3, line 114 – do the authors mean three-dimensional structure at the atomic level (e.g., crystal structure)?
Michaelis constant and catalytic efficiency should be indicated as KM (capital “M” bottom index) and kcat (“k” is lower case latter, “cat” bottom index).

Author Response

REVIEWER 2:

PONs are an essential group of enzymes involved in the metabolism of various substrates, including organophosphates, aryl esters, and some lactones. In this review, the authors focus mainly on PON3 members of paraoxonases and describe their specific activities and their role in various diseases such as cancers, HIV, and CVD. The topic is generally described in detail and seems technically sound, and the results are of general interest. The manuscript is, in principle, suitable for publication if the issues below are adequately addressed in a revised version.

MAJOR comment:

  • Because the review may be of general interest, I suggest comparing PONs members at a molecular level (amino acid alignments, domain composition, etc.). Are there any differences between the glycosylation profile of paraoxonases among PON1-3?

Reply: Thank you for this suggestion, we agree and have now included additional discussion comparing PONs members at a molecular level (amino acid alignments, domain composition, etc.) as well as differences between the glycosylation profile of paraoxonases among PON1-3 and included additional references, as well as a supplemental table.

Page 2, line 51: the sentence: “…they exhibit almost no true “paraoxonase” activity…” is unclear. Could the authors explain that in a more detailed manner?

Reply: Thank you for this suggestion, we have now clarified this sentence to indicate that “…they exhibit almost no true “paraoxonase” activity [i.e. hydrolytic activity toward paraoxon (diethyl p-nitrophenyl phosphate) as a substrate ]”

  • Page 2, subsection 1.1.2: there is confusion, as the authors indicate that PON3 has low arylesterase activity compared to PON1 (lines 80-81), however in line 84, they tell something different. 

Reply: Thank you for this clarification, we have now edited this subsection to read “While all three PON isoforms exhibit arylesterase activity, it is very low for PON2 [29,30].  Phenyl acetate is one of the best-known aryl ester substrates for PON1 but is hydrolyzed at a modest rate by PON3 and very slowly by PON2 [3,4,29].”

  • Page 3, line 114 – do the authors mean three-dimensional structure at the atomic level (e.g., crystal structure)? Michaelis constant and catalytic efficiency should be indicated as KM (capital “M” bottom index) and kcat (“k” is lower case latter, “cat” bottom index).

 Reply: Thank you for this clarification, we have now specified this sentence to indicate that we are referring to the crystal structure and we have fixed the KM and kcat references.